# Phi-Value and NMR Structural Analysis of a Coupled Native-State Prolyl Isomerization and Conformational Protein Folding Process

**DOI:** 10.3390/biom15020259

**Published:** 2025-02-10

**Authors:** Ulrich Weininger, Maximilian von Delbrück, Franz X. Schmid, Roman P. Jakob

**Affiliations:** 1Institute of Physics, Biophysics, Martin-Luther-University Halle-Wittenberg, 06120 Halle (Saale), Germany; ulrich.weininger@physik.uni-halle.de; 2Laboratorium für Biochemie und Bayreuther Zentrum für Molekulare Biowissenschaften, Universität Bayreuth, 95447 Bayreuth, Germany; delbrueck@knauer.net (M.v.D.); fx.schmid@uni-bayreuth.de (F.X.S.); 3Focal Area Structural Biology, Biozentrum, University of Basel, Spitalstrasse 41, 4056 Basel, Switzerland

**Keywords:** protein folding, prolyl isomerization, NMR spectroscopy, Phi-value analysis, protein stability

## Abstract

Prolyl *cis*/*trans* isomerization is a rate-limiting step in protein folding, often coupling directly to the acquisition of native structure. Here, we investigated the interplay between folding and prolyl isomerization in the N2 domain of the gene-3-protein from filamentous phage fd, which adopts a native-state *cis*/*trans* equilibrium at Pro161. Using mutational and Φ-value analysis, we identified a discrete folding nucleus encompassing the β-strands surrounding Pro161. These native-like interactions form early in the folding pathway and provide the energy to shift the *cis/trans* equilibrium toward the *cis* form. Variations distant from the Pro161-loop have minimal impact on the *cis*/*trans* ratio, underscoring the spatial specificity and localized control of the isomerization process. Using NMR spectroscopy, we determined the structures for both native N2 forms. The *cis*- and *trans*-Pro161 conformations are overall identical and exhibit only slight differences around the Pro161-loop. The *cis*-conformation adopts a more compact structure with improved backbone hydrogen bonding, explaining the approximately 10 kJ·mol^−1^ stability increase of the *cis* state. Our findings highlight that prolyl isomerization in the N2 domain is governed by a localized folding nucleus rather than global stability changes. This localized energetic coupling ensures that proline isomerization is not simply a passive, slow step but an integral component of the folding landscape, optimizing both the formation of native structure and the establishment of the *cis*-conformation.

## 1. Introduction

The *cis*/*trans* isomerization of peptidyl–prolyl bonds in proteins is inherently slow [1,2], with the *trans* form generally being favored in unfolded or newly synthesized polypeptide chains [3]. Proteins that contain *cis* prolyl bonds in their native state must undergo a *trans*-to-*cis* isomerization during folding. This isomerization process is closely linked to conformational folding, where initial folding often begins with certain proline residues in the incorrect (*trans*) state. These *trans*-proline residues create a kinetic barrier, temporarily halting folding and allowing conformational energy to accumulate [4,5,6,7]. This accumulated strain then drives the equilibrium toward the native *cis* form. After proline isomerization, folding can proceed rapidly to completion.

In the folded state, the *cis* isomer is favored because it allows for stronger stabilizing interactions compared to the *trans* isomer, resulting in an energetic coupling between folding and prolyl isomerization. The folded structure stabilizes the *cis* isomer, while the *cis* form further enhances the stability of the protein through additional interactions. During folding, this cooperative relationship evolves stepwise, as conformational energy initially drives the shift in the *cis*/*trans* equilibrium, but the stabilizing interactions of the native state only form after the slow *trans*-to-*cis* isomerization. This represents the rate-limiting step of the folding process and at the same time locks in the native state [8,9,10].

Prolyl isomerization not only serves as a critical rate-limiting step in protein folding but also functions as a molecular switch or timer in various biological processes [11,12,13,14,15]. Prolyl isomerases—enzymes that accelerate these isomerization reactions—are widely distributed and play essential roles in cellular function [16,17,18].

Studying folding intermediates or misfolded states containing non-native prolyl isomers remains challenging due to their instability and low population levels. These species are typically difficult to characterize structurally under equilibrium conditions, although, in rare cases, the coexistence of *cis* and *trans* isomers has been detected using NMR spectroscopy. However, the detailed structural characterization of these minor species has proven elusive [19,20,21,22,23,24,25].

The N2 domain of the gene-3-protein from the filamentous phage fd presents an ideal model system to investigate the link between folding and prolyl isomerization, as it enables the simultaneous assessment of the stability and unfolding/refolding kinetics of both the *cis* and *trans* forms [26]. The N2 domain folds independently, with Pro161, located at the tip of a β hairpin (Figure 1A), existing as a mixture of two native folded states differing in the *cis*/*trans* conformation of Pro161. During refolding, the *cis* content increases from 7% in the unfolded state to 90% in the folded state [26]. Both *cis* and *trans* forms of N2 represent true native states, displaying identical unfolding rates but differing in refolding kinetics. This difference suggests that the conformational energy driving *cis*/*trans* isomerization is already available in the folding transition state. The relationship between folding and prolyl isomerization in both the unfolded and folded states of N2 is effectively illustrated by the box model in Figure 1B, which links conformational folding with prolyl isomerization.

In our previous work, we used single- and double-mixing kinetic experiments, alongside mutational analysis, to determine the source of energy that shifts the *cis*/*trans* ratio of Pro161. We found that this energy largely originates from the two-stranded β sheet at the base of the Pro161 hairpin [27].

Building on this foundation, the current study expands the mutational analysis to 35 additional residues across the N2 domain. We use Φ-value analysis [28,29,30,31,32,33] to investigate the effects of each mutation on folding kinetics and stability, thereby gaining insights into the folding pathway of the N2 domain. Additionally, we used NMR spectroscopy to determine the solution structures of the *cis* and *trans* forms of Pro161, elucidating the molecular basis for their differing stabilities.
Figure 1(**A**) Tertiary structure of the N2 domain of G3P (residues 102–205). The side chains of His129, Pro161, Trp181, and the nine Tyr residues are shown in stick representation. Trp181 is located behind helix1. The figure was prepared using PyMol [34] and the crystal structure of full-length G3P (PDB ID: 1G3P) [35]. (**B**) Model for the coupling between folding and prolyl isomerization of the N2 domain. (**C**) Amino acid sequence of the N2 domain. Pro161 is shown in bold, in blue are shown amino acid positions analyzed before [27], and amino acid positions analyzed in this work are colored green.
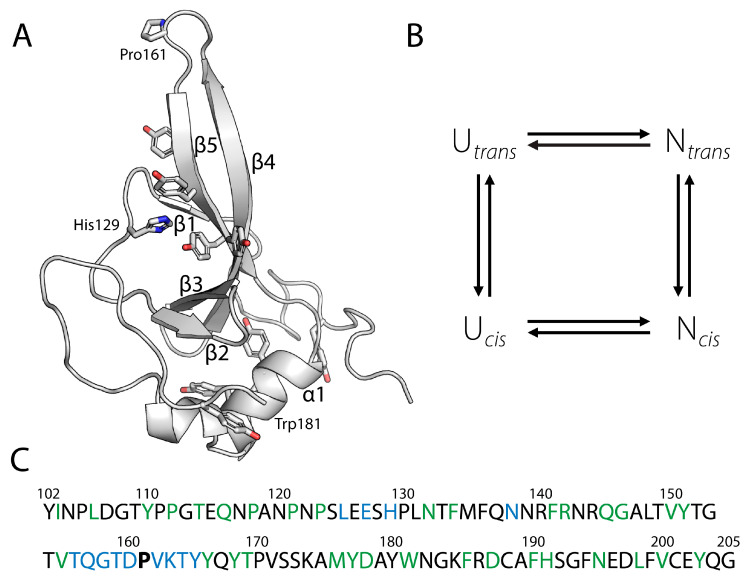


## 2. Materials and Methods

### 2.1. Mutagenesis, Protein Purification and Sample Preparation

The isolated N2 domain [residues 102–205 of the gene-3-protein of phage fd, extended by (His)6] with the stabilizing mutation Q129H was used as the reference (pseudo-wild-type) protein. The site-directed mutagenesis of N2′ was performed by BluntEnd-PCR [36] based on the expression plasmid pET11a (Novagen, Madison, WI, USA) The proteins were overproduced as inclusion bodies in *E. coli* BL21(DE3) and purified as described previously [26]. The concentrations of the N2-variants were determined via the absorption and the molar extinction coefficient ε_280_ = 19,000 M^−1^cm^−1^ at 280 nm. All buffers for spectroscopic measurements were dust filtered through 0.22 μm nylon filters before use and degassed in the desiccator with a membrane vacuum pump. The protein stock solutions were thawed at 4 °C, and any aggregates present were removed by centrifugation (approx. 30 min, 4 °C, 13,000 rpm, laboratory centrifuge).

### 2.2. Measurement of Near-UV Far-UV CD Spectra

All Circular dichroism spectra were measured using a Jasco J-600 (Tokyo, Japan) spectropolarimeter. Near-UV circular dichroism spectra (260–320 nm) were recorded with a protein concentration of 50 μM in 100 mM potassium phosphate, pH 7.0, in temperature-controlled 10 mm cuvettes. The step size was 0.2 nm at a measuring speed of 100 nm/min, the bandwidth was 2 nm, and the attenuation was 2 s. Spectra were measured ten times, averaged, and corrected for the contribution of the buffer. Far-UV spectra (185–250 nm) were measured in 10 mM potassium phosphate, pH 7.0, and a protein concentration of 5 µM.

### 2.3. Equilibrium Unfolding Transitions

For urea-induced unfolding, protein samples (1.0 μM) were incubated for 1 h at 15 °C in 100 mM K-phosphate, pH 7.0, and different concentrations of urea. The sample fluorescence was measured in 1 cm cuvettes at 340 nm (10 nm bandwidth) after excitation at 280 nm (5 nm bandwidth) (Hitachi F4010 fluorescence spectrometer). The data were analyzed according to a two-state model by assuming that Δ*G*_D_ as well as the fluorescence emissions of the folded and unfolded forms depend linearly on the urea concentration. A nonlinear least squares fit with proportional weighting of the experimental data was used to obtain Δ*G*_D_ as a function of urea concentration [37]. The heat-induced unfolding transitions were recorded in a Jasco J-600 spectropolarimeter equipped with a PTC 348 WI Peltier element at a protein concentration of 4 μM in 100 mM K-phosphate, pH 7.0 at a heating rate of 1 °C/min. The transitions were monitored by the CD signal increase at 222 nm with 1 nm bandwidth and 10 mm path length. The experimental data were analyzed based on the two-state approximation, with a heat capacity change ΔCp of 6400 J mol^−1^ K^−1^ [38].

### 2.4. Kinetic Experiments

All urea-induced unfolding and refolding experiments were performed in 100 mM K phosphate, pH 7.0, at 15 °C at a final protein concentration of 0.5 μM using a DX.17MV stopped-flow spectrometer from Applied Photophysics (Leatherhead, UK). The native or unfolded (in 4.4 M urea) protein was diluted 11-fold with urea solutions of different concentrations. The kinetics were monitored by the change in fluorescence over 320 nm after excitation at 280 nm (10 nm bandwidth) in an observation cell with a 2 mm path length. A 0.5 cm cell containing acetone was placed between the observation chamber and the photomultiplier to absorb scattered light from the excitation beam. The kinetics were measured at least eight times under identical conditions and averaged to improve the signal-to-noise ratio. In the analysis of the unfolding and refolding kinetics of the individual variants, we assumed that the folding kinetics of the *cis* and the *trans* forms are kinetically isolated by the slow U_t_/U_c_ and N_t_/N_c_ isomerizations and that the logarithms of the microscopic rate constants of unfolding and refolding depend linearly on the urea concentration. The Δ*G*_UN_ values were determined from the ratio of the rate constants for refolding and unfolding [Δ*G*_UN_ = −*RT*ln(*k*_UN_/*k*_NU_)]. ΔΔ*G*_UN_ is the difference between the Δ*G*_UN_ values of the mutant and the wild-type protein. The ΔΔ*G*_UN_^‡^ values were derived from the ratio of the refolding rate constants of the mutant (mt) and the wild-type protein (wt) [ΔΔ*G*_UN_^‡^ = −*RT*ln(*k*_UN_(mt)/*k*_UN_(wt))], and Φ is the ΔΔ*G*_UN_^‡^/ΔΔ*G*_UN_ ratio [39]. In order to keep the error caused by extrapolation as low as possible, the Φ-values for the unfolding were calculated for 2 M urea.

Interrupted unfolding experiments using double-mixing stopped-flow techniques were conducted to determine the *cis*/*trans* ratio in the folded forms of the N2′ variants at 15 °C. In the first step, 33 µM of folded N2′ protein (in 100 mM K phosphate, pH 7.0) was diluted 11-fold with 100 mM glycine buffer, initiating unfolding at a final pH of 2.0. Under these conditions, complete conformational unfolding occurred within 10 ms, while Pro161 *cis*/*trans* equilibration exhibited a time constant of 55 s [26]. After 100 ms, the N2′ variants were fully unfolded, but the *cis*/*trans* ratio remained virtually identical to that in the folded protein. Refolding was triggered after a 100 ms delay by an additional six-fold dilution, resulting in a final protein concentration of 0.5 µM in 100 mM K phosphate, pH 7.0. The *cis* content in the folded protein was determined by calculating the amplitude ratio of refolding reactions corresponding to the *cis* and *trans* isomers, considering slight fluorescence differences between N_c_ and N_t_. Each experiment was repeated 10 times for all variants. Individual measurements were analyzed separately, with *cis* content variability remaining within 3%.

The Gibbs free energy that is necessary to shift the *cis*/*trans* equilibrium, referred to as the proline shift energy, is given by the equation −*RT*ln(*K*_N_/*K*_U_). Here, *K*_N_ and *K*_U_ represent the measured equilibrium constants for Pro161 *cis*/*trans* isomerization in the unfolded state (*K*_U_ = [U_t_]/[U_c_]) and in the native state (*K*_N_ = [N_t_]/[N_c_]), respectively.

### 2.5. NMR Spectroscopy

All NMR experiments were performed on a Bruker Avance II 600 spectrometer at 15 °C in 100 mM K phosphate buffer and 10% (*v/v*) D_2_O at pH 7.0. Spectra were processed using NMRPipe [40] and analyzed using NMRView [41]. Backbone resonances have been assigned previously [42]. Aliphatic side chains were assigned by H(C)CH-TOCSY [43], and side chain NH and aromatic side chains by NOEs. NOEs for the structure determination were derived from 3D-NOESY-HSQC experiments for ^15^N and ^13^C aliphatic nuclei and a 2D NOESY experiment. Phi–Psi dihedral angle constraints were derived using TALOS [44]. H-bonds were introduced if all the following criteria were fulfilled: amide exchange of the corresponding amide is slowed down [42], NOE patterns are in agreement with H-bonds, they are located in secondary structure elements confirmed by chemical shifts and initial structures. ^1^H/^15^N RDCs were determined in 18 mg/mL PF1 phages from PROFOS. RDCs were used for isolated amide signals with ^1^H^15^N NOEs > 0.6. RDCs are located all over the structure. Structure calculations were performed using ARIA 2.3 [45]. 50 structures have been calculated using NOEs as ambiguous distance restraints, above mentioned Phi–Psi dihedral angle constraints, H-bonds and RDCs, and standard ARIA parameters for each Pro161 in *cis* or *trans*. Pro161 has been confirmed as *cis* in the NMR spectra; its surrounding was determined as flexible by ^1^H^15^N NOE experiments [42]. The *trans* structure was calculated with the same data set.

## 3. Results

### 3.1. Design of the Protein Variants

We employed protein engineering and Φ-value analysis [30,31,46] to investigate the transition state of folding for the N2 domain of the phage gene-3-protein. A total of 35 single-point mutations were introduced into the protein, and for each mutant, we measured the differences in Gibbs free energy of folding (ΔΔ*G*_UN_), mutant minus wild-type protein, and activation free energy of refolding (ΔΔ*G*_UN_^‡^). These values were derived from equilibrium unfolding transitions and folding kinetics, respectively. The Φ-value, calculated as the ratio of these two free energies, was determined for both the *trans* and *cis* forms of Pro161:Φ = ΔΔ*G*_UN_^‡^/ΔΔ*G*_UN_

A Φ-value of 1 indicates that the mutated side-chain is in a native-like environment in the transition state, contributing equally to the stability of both the native state and the transition state. In such cases, only the refolding kinetics are affected. Conversely, a Φ-value of 0 signifies that the side-chain is in an unfolded-like region of the transition state, affecting only the unfolding kinetics. Fractional Φ-values suggest partial structural formation at the mutation site or the presence of multiple folding pathways.

The N2 domain of the phage fd gene-3-protein, comprising residues 102–205 (Figure 1A), is only marginally stable in isolation. Its stability is enhanced by approximately 8 kJ·mol^−1^ in Gibbs free energy of denaturation (Δ*G*_D_) by the Q129H mutation, identified via an in vitro selection [47]. For this study, we used the Q129H variant as our pseudo-wild-type protein, referred to as N2′.

Single mutations selected for the Φ-value analysis were evenly distributed over the protein (Figure 1C), except for regions previously analyzed (aa125–129; aa 136–139; aa 156–165,) [27]. With the exception of L198P, which was identified previously through in vitro selection [48], all residues were substituted by alanine. All N2′-variants were expressed in *E. coli* BL21 as inclusion bodies, refolded, and purified via Ni-affinity and size-exclusion chromatographies. Most variants exhibited lower refolding yields compared to the wild-type protein (Figure 2A). Five variants (F134A, F141A, Y168A, V171A, Y177A) aggregated after refolding, and two (V150A, Y151A) yielded only 1 mg. These seven residues are hydrophobic and located in the N2′ core, where their interactions are critical for protein stability (Figure 2B).

CD spectroscopy was used as a quick and effective method for analyzing the secondary and tertiary structure of N2′ variants. Checking their correct folding is particularly important for Φ-value analysis, which is based on data from highly destabilized variants.

Far-UV CD spectra (185–250 nm) of the N2′ domain, consisting of five β-strands and a short α-helix, displays a β-sheet characteristic spectrum with a maximum at 194 nm and a minimum at 218 nm (Appendix A). The positive signal between 190 nm and 200 nm confirms the folded state of N2′ variants. The virtual identity of the far-UV CD spectra of N2′ and the variants (P112A, N132A, M176A, D187A, V200A) demonstrates that the substitutions to alanine did not affect the backbone structure of the folded variants.

The near-UV CD spectrum (260–320 nm) reflects the asymmetric immobilization of aromatic groups and thus the tertiary structure. N2′ shows a structural “fingerprint” with a maximum at 284 nm and a minimum at 266 nm (Appendix A), and again the variant proteins show the same spectra as N2′. This confirms that the substitutions to Ala also did not affect the tertiary structure of N2′.

### 3.2. Stabilities of the Protein Variants

The effect of alanine substitutions on the conformational stability of the N2′ domain was investigated by following thermal unfolding transitions and urea-induced equilibrium transitions. Thermal unfolding was monitored by measuring the ellipticity at 222 nm (Appendix A) and analyzed using a two-state model. To ensure comparability, the transitions were normalized to the fraction of native protein. Appendix A summarizes the normalized thermally induced transitions for all N2′ variants, grouped by increasing stability (A to F). Gibbs free energies of thermal unfolding (Δ*G*_D_) were extrapolated to 32 °C, the average midpoint temperature of all N2′ variants, to minimize extrapolation errors. Free energies were also calculated at 15 °C to correlate with thermodynamic stabilities derived from urea-induced transitions (Appendix A). For the highly destabilized variants L106A and V150A, the poorly defined baseline slopes of the native protein were adjusted using data from more stable variants. The N2′ variants displayed a wide range of mutation effects on stability, with most variants showing stability changes within ±5 kJ·mol^−1^.

Urea-induced unfolding transitions of N2′ variants were measured using tryptophan emission at 340 nm (excitation at 280 nm) (Appendix A). Most variants showed fluorescence properties similar to the wild-type, except N2′-Y110A and N2′-W181A. The N2′ domain contains nine tyrosines and one tryptophan (W181) (Figure 1A), facilitating Förster Resonance Energy Transfer (FRET). The Tyrosine mutation Y110A reduced fluorescence by 60%, replacing W181 with alanine eliminated FRET, causing tyrosine fluorescence to increase due to environmental changes. Data were analyzed with a two-state model and normalized to native protein fractions (Appendix A). Analysis of highly destabilized variants required fixed baselines. Thermodynamic parameters (Appendix A) showed cooperativity across variants, similar to the wild-type protein. ΔG values at 15 °C in 2 M urea were extrapolated for Φ-value calculations.

Stabilities from thermal and urea-induced unfolding transitions correlate well, as shown by the alignment of midpoints [urea]_M_ and *T*_M_ in Figure 3A. Figure 3B shows the impact of alanine substitutions on N2′ stability (ΔΔ*G*_D_). Destabilizing mutations are distributed throughout the domain. Strong destabilization at the protein N-terminus is followed by a region with reduced destabilization (residues 116–145). This less destabilized region encompasses the loops that precede and extend from β-strand 2 and 3 (Figure 1A), and it is followed by another region with strong destabilizing mutations (residues 150–190). Even residues near the C-terminus (V200, Y203) make significant contributions to protein stability.

### 3.3. Folding Kinetics of the Protein Variants

In stopped-flow experiments, the unfolding and refolding kinetics of the N2′ variants were measured by fluorescence as a function of the urea concentration. In the denatured state, N2′ exists as 93% *trans-* and 7% *cis*-Pro161 conformers [26], differing in stability and causing biexponential refolding kinetics. The main refolding phase (*trans*-conformer) has a large amplitude. The *cis*-conformer refolds more rapidly with a smaller amplitude. Slow proline isomerization (τ~100 s) is decoupled from folding and not analyzed further.

In contrast to biphasic refolding, N2′ unfolding follows a monoexponential course, as both conformers unfold at the same rate. Appendix A summarize the rates of all N2′ variants from refolding and unfolding experiments as a function of urea concentration, plotted in a semi-logarithmic fashion in so-called Chevron plots. For comparison with the wild-type protein, the fit to N2′ wild-type data (in red) is included in each Chevron plot.

The results of the Chevron analyses, based on a two-state model, are listed in Appendix A. All values are calculated at 2 M urea to minimize extrapolation errors. The unfolding arms of all N2′ variants were well-defined and analyzable. However, due to the shift of the Chevron plot to lower urea concentrations, the refolding arms of the *trans*-conformers of several strongly destabilized variants (I103A, L106A, Y110A, G146A, V150A, Y151A, G153A, M176A, F185A, D187A) could not be analyzed. The dependence of apparent rates on urea concentration is similar for the wild-type protein and N2′ variants, with nearly identical unfolding slopes producing comparable Chevron plot shapes. The largest differences occur in the unfolding arms, where substitutions often accelerate unfolding, and can also be seen in the small changes in free activation enthalpy (ΔΔ*G*_UN_^‡^) for refolding (Figure 4A) and large enthalpy changes for unfolding (ΔΔ*G*_NU_^‡^) (Figure 4B). The average β-Tanford value (*β*_T_) of 0.66 for the N2′ variants indicates that approximately two-thirds of the hydrophobic interior surface is buried in the transition state already (Appendix A). The similar *β*_T_ values observed across all N2′ variants suggest that replacing individual amino acids did not alter the overall packing of the protein interior in the transition state.

The sum of ΔΔ*G*_UN_^‡^ and ΔΔ*G*_NU_^‡^ corresponds to the total destabilization caused by the amino acid substitution and indeed matches very well with the Gibbs free energy differences obtained from the equilibrium transitions, ΔΔG_D_, for most N2′ variants. Figure 4C illustrates this by plotting these ΔΔG_D_ values against the sum of the activation parameters, ΔΔ*G*_UN_^‡^ + ΔΔ*G*_NU_^‡^, for the *cis*-conformer from the refolding and unfolding experiments. The slope of the regression line, close to one, indicates a very good correlation between the kinetic and thermodynamic measurements.

Figure 4D presents a summary of all available Φ-values for the *cis*-form of the N2′ domain from this work and Jakob & Schmid (2009) [27]. Our analysis focuses on the *cis* form because, for many strongly destabilized N2′ variants, analyzing the *trans* form was either infeasible or resulted in high errors. The figure highlights a highly uneven distribution of Φ-values. Most of the N2′ domain exhibits low Φ-values (shown in blue in Figure 4D), indicating that these amino acids form their stabilizing interactions only after passing the transition state.

Distinct from this general trend is the Pro161-loop, where high Φ-values reported by Jakob & Schmid (2009) are confirmed by elevated Φ-values in the previous and subsequent Beta-strand (e.g., V155A, Y166A). This region seems to establish native-like interactions very early during folding. Interestingly, two abrupt increases in Φ-values, unrelated to side-chain interactions, were caused by backbone mutations. In G146A and G153A, conformational restrictions due to the more rigid alanine likely hindered refolding, explaining the sharp increase in Φ.

These results demonstrate that the N2′ domain has a well-defined folding nucleus, primarily formed by the hairpin structure comprising β-strands 4 and 5 and the intervening loop around Pro161. Notably, this same region is responsible for the *cis/trans* equilibrium at Pro161.

### 3.4. Distant Variations Have No Impact on the cis/trans Ratio at Pro161

Pro161 is located in a large loop between two β-strands. Previous experiments suggested that the *cis*/*trans* ratio at Pro161 in the native protein is mainly determined by interactions between these β-strands. In the *cis*-conformation, these interactions are stronger, resulting in a more stable, native-like structure that favors the *cis*-conformer. In the presence of *trans*-Pro161, the connecting peptides are locally unfolded, structurally and energetically decoupling Pro161 from the β-strands (Jakob & Schmid, 2009). To measure both Uc → Nc and Ut → Nt refolding reactions with large amplitudes, a stopped-flow double-mixing protocol was employed.

In the first step, native N2′ was completely unfolded by a short 0.1 s long pH jump from 7.0 to 2.0 and then refolded in the initial buffer. Since the *cis*/*trans* ratio remained virtually unchanged during the 0.1 s unfolding pulse, the refolding amplitudes reflect the native *cis*/*trans* ratio at Pro161. The very small signal change from Nt ⇌ Nc equilibrium adjustment was taken into account when calculating the actual N_t_ content from the U_t_ → N_t_ amplitude [26]. The results from the double-mixing experiments clearly show that mutations outside the two β-strands have little impact on the equilibrium between the two conformers (Figure 5). The *cis*-fractions varied within a narrow range between 86% and 94%. The most significant reduction, down to 80%, was observed for the V155A substitution (Figure 5). V155A is located in the β-strand leading to the Pro161-loop (residues 157–164).

These double-mixing experiments demonstrate that global stability changes have no effect on the conformer ratio. Most studied positions are far from the exposed loop containing Pro161, supporting the idea that the *cis*/*trans* ratio is determined locally by interactions between the β-strands leading into the Pro161 loop and not by changes in the overall stability of N2′.

### 3.5. NMR Structure Determination of cis and trans-Form of N2′

Using Phi-value analysis, we found that the two β strands flanking Pro161 form the folding nucleus of the N2′ domain. This same region determines the native-state *cis*/*trans* ratio at Pro161. To obtain a structural model for both *cis-* and *trans-* forms of the N2 domain and explain their distinct stabilities, we used solution NMR spectroscopy. The N2′ backbone was previously assigned [42], and we have now assigned the side chain resonances as well. NMR spectra generally showed a single set of peaks (e.g., ^1^H-^15^N HSQC in Appendix A), indicating that we have just one main NMR state, with only a few weak additional signals, often from prolines themselves. These few weak signals suggest that, in the folded state, each proline primarily adopts either a single *cis* or *trans* conformation or, if a minor conformation exists, it has only a limited local impact on the structure.

Heteronuclear NOE (hNOE) measurements revealed that the Pro161-loop region exhibits notably low hNOE values [42]. Low hNOE values are indicative of elevated backbone flexibility and increased local mobility on the ps–ns timescale. This finding implies that, while the Pro161-loop forms the folding nucleus of the N2′ domain, it is less rigid and more dynamic than the neighbouring regions, suggesting that structural differences in the *trans* and *cis* forms of Pro are restricted to the β-sheets and loop directly nearby Pro161. In addition, differences in the β-sheets might be averaged out in the NMR spectra.

Of the nine prolines in N2′, NMR data clearly define only two states: Pro118 and Pro130 are in *trans*-state. For the NMR structure determination of both states, we treated Pro161 as *cis* or *trans* and all others as *trans*, based on available crystal structures [48]. Under these assumptions, we obtained a well-folded structural ensemble consistent with all NMR data (Appendix A). The structural ensembles are well-defined and only show deviations close to the protein termini and loop regions (Appendix A). With an r.s.m.d. of 1.5 Å, the NMR model of the isolated N2′ closely matches the X-ray structure (PDB ID: 1G3P) [35] of the N2 domain in full-length gene-3-protein (Figure 6A). Minor differences are restricted to loop regions, where low hNOE values confirm increased flexibility [42]. Thus, the isolated N2 domain exhibits a very similar fold as within the full-length protein. In addition, we calculated a structure for *trans*-Pro161. This structure uses the same NMR data as the *cis*-Pro161 structure, since we were unable to obtain any pure *trans*-Pro161 NMR data. The only difference is the actual conformation of Pro161. As expected by such an approach and considerations above, both *cis*-Pro161 and *trans*-Pro161 NMR structures are virtually identical, differing only slightly in the β-strands near Pro161 (Figure 6B). Both *cis* and *trans* models fit the NMR data equally well and exhibit nearly identical energies in the structural refinements (see Appendix A), indicating that structural differences are limited to the β-sheets and loops surrounding Pro161, and neither the absence of only *trans*-Pro161 NMR data nor the use of potentially only *cis*-Pro161 NMR data has a sizeable impact on the structures.

In the trans-Pro161 conformation, the Pro161-loop adopts a more extended structure (Figure 6C), resulting in four backbone hydrogen bonds forming between residues 155 and 165. In contrast, when Pro161 is *cis*, the loop becomes more compact. This compaction shortens the Gly158–Lys163 backbone hydrogen bond (3.0 Å in *cis* vs. 3.4 Å in *trans*) and introduces an additional backbone hydrogen bond (Thr159–Lys163) (Figure 6D). Together, these improved hydrogen-bonding interactions likely account for the approximately 10 kJ·mol^−1^ higher stability of the *cis* conformation.

## 4. Discussion

Our results highlight the intricate balance between folding and prolyl isomerization in the N2 domain of the gene-3-protein and underscore the importance of localized interactions in determining the *cis*/*trans* equilibrium at Pro161. In the N2 domain, Pro161 is a critical residue where the *cis* isomer dominates the native state, despite the trans isomer being strongly favored in the unfolded ensemble [31,32]. Our expanded mutational analysis and subsequent Φ-value measurements reveal that the early formation of native-like interactions in the β-sheet region encompassing Pro161 is the central driver of this shift. These data support a model in which the energetic coupling between folding and prolyl isomerization is localized rather than global, allowing the N2 domain to accumulate the conformational energy required to flip the Pro161 peptide bond as folding progresses.

The Φ-value distribution (Figure 4D) shows that most residues form their native-like stabilizing interactions only after the folding transition state has been crossed, consistent with a cooperative, two-state folding mechanism. In contrast, the β-sheets surrounding Pro161 emerge as a clear “folding nucleus”, as also found for other small proteins [28,49,50,51,52,53,54,55]. High Φ-values in the Pro161-loop region, as well as the heightened sensitivity to local substitutions (e.g., V155A) that alter the *cis*/*trans* ratio, demonstrate that these local β-strand interactions are established early and guide the protein toward the native *cis*-conformation. The lack of significant changes in the *cis*/*trans* ratio for mutations outside the immediate vicinity of Pro161 further reinforces the idea that the *cis*/*trans* equilibrium and its shift during folding is governed largely by local interactions (Figure 5). Alterations in stability or folding kinetics remote from the Pro161-loop do not substantially shift the *cis*/*trans* ratio. This points to a remarkable local spatial specificity for the mechanism of coupling between prolyl isomerization and conformational changes in the protein. Native-state prolyl isomerizations used as molecular switches are also controlled by precise local interactions to regulate communications pathways [15,16,56,57,58,59,60].

Our NMR structural studies corroborate these findings by demonstrating that the *cis* and *trans* conformations differ only slightly, primarily in backbone hydrogen bonding around Pro161. Both forms exhibit nearly identical global folds, implying no drastic structural rearrangements outside this local region (Figure 6). The more compact *cis* form of the Pro161-loop is stabilized by an additional hydrogen bond and a shortening of existing hydrogen bonds. Both factors contribute probably to the 10 kJ·mol^−1^ increase in stability. These subtle and localized changes confirm that the energetic and structural determinants of prolyl isomerization are localized to a narrowly confined region around Pro161. Thus, the NMR data link the mechanistic picture from kinetic and thermodynamic analyses to tangible structural features, illustrating how minimal structural rearrangements can yield significant energetic differences between *cis* and *trans* states.

In summary, our combined kinetic, thermodynamic, and structural results reveal that the *cis*/*trans* equilibrium at Pro161 is tightly intertwined with the early formation of native-like β-sheet interactions that shape the protein’s folding landscape. This interplay ensures that prolyl isomerization is not a mere passive hurdle but an integral, controlled step within the folding pathway.

## 5. Conclusions

Mutational and Φ-value analyses pinpoint the β-strands surrounding Pro161 in the N2 domain as critical for both early protein folding and shifting the *cis*/trans equilibrium. Substitutions in distant regions have little effect on the *cis*/*trans* ratio of Pro161, underscoring the spatial specificity of this coupling. NMR structures of the *cis* and *trans* forms reveal nearly identical global folds, yet subtle differences in backbone hydrogen bonding around Pro161 provide an explanation for the higher thermodynamic stability of the *cis* conformation. Together, these results show that proline isomerization and the native-state Pro161 *cis*/*trans* equilibrium are integral parts of the protein folding mechanism, governed by local energetic and structural features in the protein.

## Figures and Tables

**Figure 2 biomolecules-15-00259-f002:**
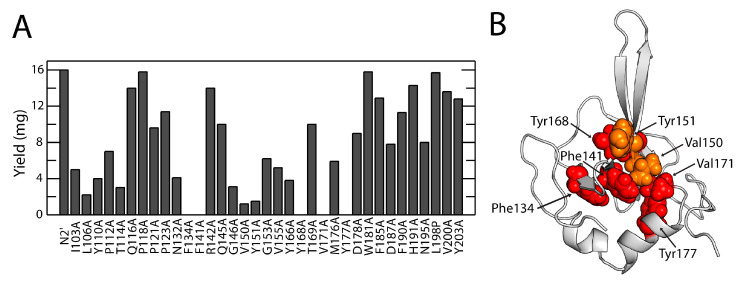
N2′ variant production (**A**) Yields (mg) of the individual N2′ variants per 2L fermentation. The 35 N2′ variants with the corresponding substitutions are given. (**B**) Protein variants with no or very low protein yields. When these amino acids were replaced with alanine, the respective variants gave very low protein yields (Val150, Tyr151, shown in orange) or precipitated (Phe134, Phe141, Tyr168, Val171, Tyr177, shown in red). These amino acids are part of the hydrophobic core of the N2′ domain and essential for stability.

**Figure 3 biomolecules-15-00259-f003:**
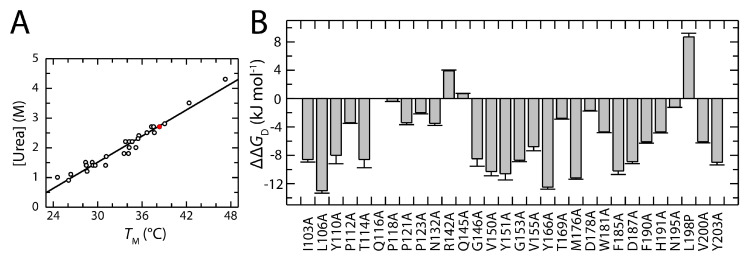
Stability analysis of the N2′ variants. (**A**) Comparison of the mid points of unfolding the N2′ variants (open dots) and the wild-type protein (filled red dot) from the thermally and urea-induced equilibrium transitions. The data shown are taken from Appendix A and were evaluated using linear regression (filled line). (**B**) Free enthalpy difference ΔΔ*G*_D_^15°C^ at 2 M urea of the individual N2′ variants compared to the wild-type protein.

**Figure 4 biomolecules-15-00259-f004:**
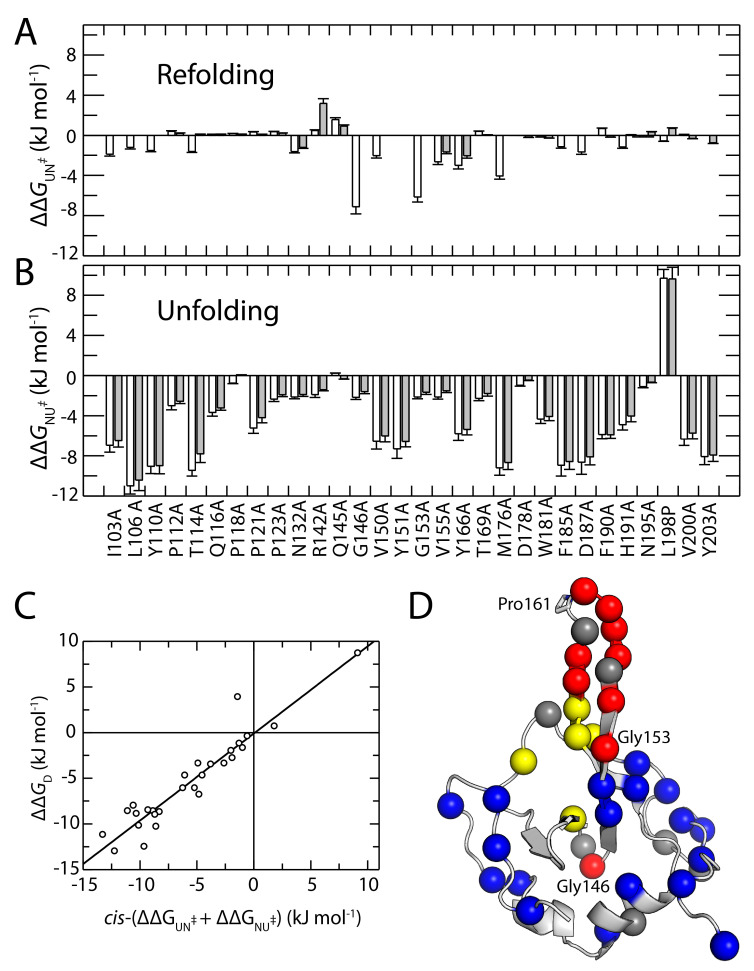
Comparison of the changes in the activation energy of refolding ΔΔ*G*_UN_^‡^, (**A**) and activation energy of unfolding ΔΔ*G*_NU_^‡^, (**B**) for the *trans* (gray) and the *cis* forms (white) of the variants. The ΔΔ*G*_NU_^‡^ and ΔΔ*G*_UN_^‡^ values are taken from Appendix A. The correlation of Gibbs free energy differences derived from kinetic measurements of the *cis*-conformer and equilibrium transitions is shown in (**C**). The regression line fitted to the data points follows the equation: ΔΔ*G*_D_ = (ΔΔ*G*_UN_^‡^ + ΔΔ*G*_NU_^‡^) · 0.95 kJ·mol^−1^ M^−1^–0.07 kJ·mol^−1^ M^−1^. All differences in Gibbs free energy are calculated for 2 M urea (at 15 °C). (**D**) The Cα-atom for amino acid positions analyzed within a Φ-value analysis are shown as spheres, including this work and previous analysis [27]. For simplification, only the Φ-values for the *cis*-Pro161 form are shown. The residues are color-coded according to their Φ-values: blue, 0.0 < Φ < 0.3; yellow, 0.3 < Φ < 0.7; red, 0.7 < Φ < 1.0; dark grey, residues with a ΔΔG_D_ < 2.0 kJ·mol^−1^.

**Figure 5 biomolecules-15-00259-f005:**
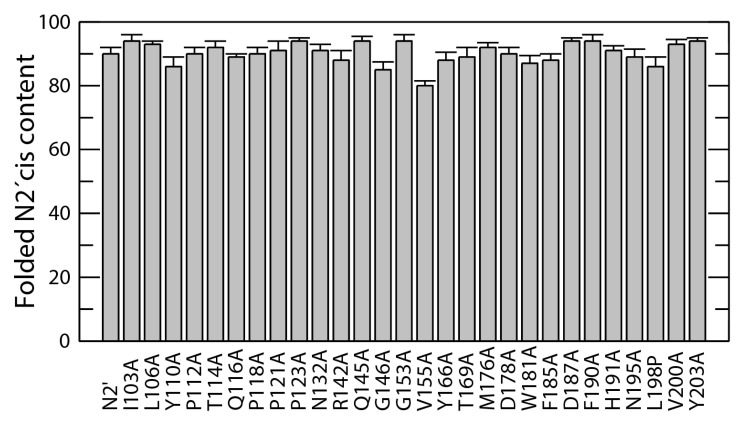
*Cis*-content at Pro161 in the folded N2′ variants. The data were determined in a double mixing experiment, as described in Materials and Methods, Section 2.4.

**Figure 6 biomolecules-15-00259-f006:**
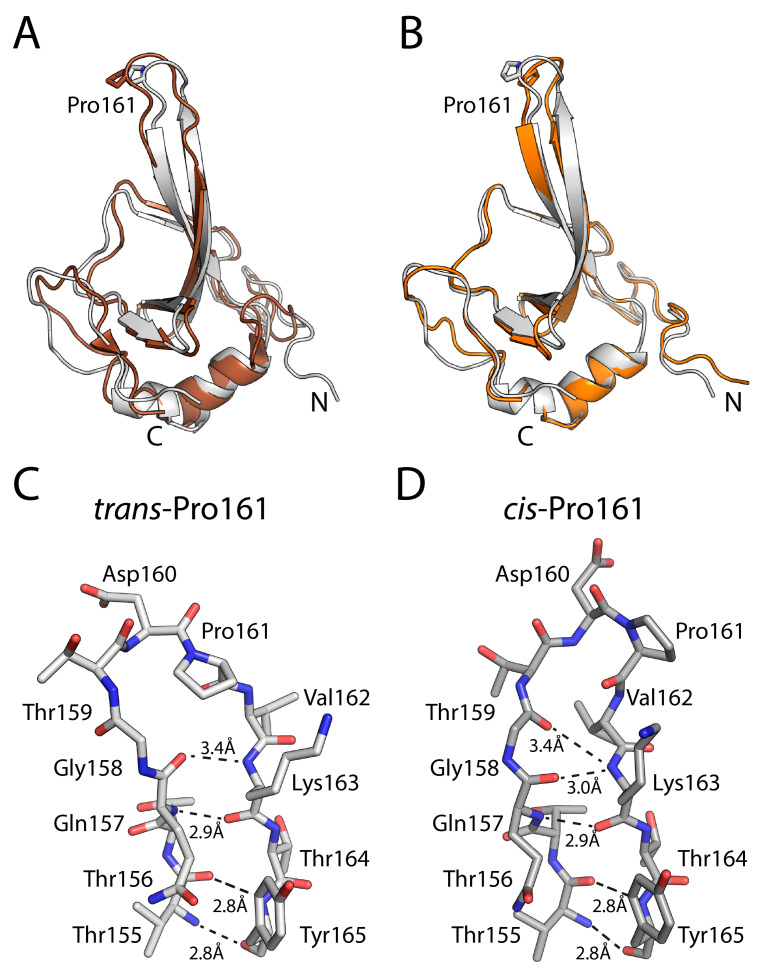
NMR structure determination of cis- and trans-conformation of N2′. (**A**) Superposition of the NMR model of N2′ containing *cis*-Pro161 (grey) with the crystal structure of the N2-domain within the gene-3-protein (PDB ID: 1G3P, brown). (**B**) Superposition of the NMR model of N2′ with cis-Pro161 (grey) and trans-Pro161 (orange). In (**A**,**B**) the N-and C-termini are indicated. Close-Up view comparison of (**C**) the *trans*-Pro (light grey, left) and (**D**) *cis*-Pro161 conformation (dark grey, right).

## Data Availability

Data are contained within the article and Appendix A.

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
