# Peer review of "Phi-Value and NMR Structural Analysis of a Coupled Native-State Prolyl Isomerization and Conformational Protein Folding Process"

_biomolecules, 2025, doi:10.3390/biom15020259_

Round 1

Reviewer 1 Report

Comments and Suggestions for Authors

The manuscript by Weininger et al highlight the intricate balance between folding and prolyl isomerization in the N2 domain of the phage gene-3 protein, underscoring the importance of localized interactions in determining the cis/trans equilibrium at Pro161. Using protein engineering and Phi-value analysis, the authors investigate the transition state ensemble of folding for this protein domain. Using NMR spectroscopy, they obtained a structural model for both cis- and trans- forms of the N2 domain, providing some explanation for their distinct stabilities.

Overall this is a good study, based on a huge amount of work, and carried out with method, using appropriate techniques, and yielding rather convincing results. I believe that the manuscript deserves publication upon addressing the following issues.

Major issues:

-  I am a little bit surprised to observe only a single population of cross-peaks in the HSQC spectrum. If the cis/trans isomerization yields two different structures, even slightly different, we should observe minor peaks corresponding to the 10% of the trans- population. The authors point out a local difference in the hydrogen network between the two conformers, at the level of the hairpin containing P161: the minor peaks should concern at least the residues involved in these different networks and should be visible in the HSQC spectra. 

-  This raises the question: can we use the NMR parameters (chemical shifts, nOes, phi/psi…) measured on the major population (corresponding to the cis- population) to model the two conformers cis- and trans- ? In my opinion: definitely not! The two populations are in slow exchange with regard to the NMR timescale (due to the energy barrier between the two isomers), and the structural information that concerns the trans- conformers should be taken from the minor peaks. Of course, with a 10% ratio when compare to the cross-peak intensities of the cis- conformer, this is not an easy task. May be the author could analyze the V155A mutant, where the trans- population is higher (20%). The other way is to suppress this section which does not bring much to the discussion anyway…

Minor Issues:

-  In the introduction, the authors state: “In the folding state, the cis isomer is favored because it allows stronger stabilizing interactions compare to the trans isomer,…” (Line 41-42). I assume that this statement concerns P161 in the N2 domain? There is no reference to this protein in the preceding sentences, so that this statement can be taken as a general statement, concerning the behavior of prolines in any all proteins. I suppose it is not?

-  There is no error-bar in the bar-diagrams. This has few importance in the case of Figure 2 (expression yields), but in the case of thermodynamics parameters (Fig.3 and 4) or populations (Fig.5), this precludes the evaluation of the relevance of the data, especially when comparing the differences between different mutants. This is particularly the case for Figure 5 where, without error-bars, it is difficult to evaluate the significance of the slight variations of the cis- population between the different mutants.

-  The authors gave few details on the NMR restraints (which residues are concerned by the measurement of RDCs? How H-bonds were determined? Etc…) and on the modeling of the protein (ARIA standard parameters is a little bit short…). I think that this should be developed in further details in the material & methods section.

-  Supp. Figure 4. It is difficult to evaluate the quality of the experimental data from the overlays of the different chevron plots (with a “c” not a capital: chevron refers to a “geometric shape” not to M. Chevron). It should be better to keep this figure with only the fits, and to present independently the individual fits (with the experimental points) in additional figures (in the Sup. Mat.)

Author Response

#Reviewer1:

Major issues:
- I am a little bit surprised to observe only a single population of cross-peaks in the HSQC spectrum. If the

cis/trans isomerization yields two different structures, even slightly different, we should observe minor peaks corresponding to the 10% of the trans- population. The authors point out a local difference in the hydrogen network between the two conformers, at the level of the hairpin containing P161: the minor peaks should concern at least the residues involved in these different networks and should be visible in the HSQC spectra.

The reviewer is correct to expect signals uniquely belonging to the trans state of Pro161. We have been expecting and looking for them as well. However, there are only very few (< 5) identifiable additional signals in the 1H15N-HSQC (the situation in the 1H13C-HSQC is similar), and none of them could be analyzed further. Furthermore, these signals can not only be addressed as the Pro161 trans state, but also as the cis states of multiple other prolines. Our conclusion therefore is that the cis/trans states of all Pro only have a very limited and local impact on the structure. In terms of Pro161 this is supported by the actual structure: Pro161 is located at the very tip of a turn connecting two beta-strands sticking out from the center of the protein. So, it is a pretty isolated region. Furthermore, this whole part is very dynamic on the ps-ns time scale (as identified by low values hNOE experiment, which has been published already, Zierer et al, J Mol Biol, 2010). Thus, differences in chemical shifts not belonging to the Pro itself, might be averaged out to some extend in the NMR spectra, as it is the case for unfolded peptides.

We have made revisions to the main text (shown in light blue) to clarify these observations and conclusions for the reader.

- This raises the question: can we use the NMR parameters (chemical shifts, NOEs, phi/psi...) measured on the major population (corresponding to the cis- population) to model the two conformers cis- and trans- ? In my opinion: definitely not! The two populations are in slow exchange with regard to the NMR timescale (due to the energy barrier between the two isomers), and the structural information that concerns the trans- conformers should be taken from the minor peaks. Of course, with a 10% ratio when compare to the cross-peak intensities of the cis- conformer, this is not an easy task. May be the author could analyze the V155A mutant, where the trans- population is higher (20%). The other way is to suppress this section which does not bring much to the discussion anyway...

We agree with the reviewer that for a clean trans structure, data from trans signals should be used. That’s what we would have done, if we could. But there are only few possible signals which we couldn’t make sense of (see above). What we do know about the trans structure is, that it should be very close to the cis structure and P161 has to be in trans. That’s why we calculated this structure anyway, in order to see what the impact of the mere existence of P161 in trans does. Any obvious imperfections of that trans structure compared to the cis structure (in terms of problematic geometry and violated NMR restraints) would point to the structural regions that are incorrect in trans. However, in all metrics the trans structure is as good as the cis structure. Can we be sure, that it is 100% correct? No! But we can be sure that it can not be far from the truth, and it is a useful structure that reproduces what we know about the Pro161 trans structure: Pro161 is in trans and the structure is very similar to the cis structure. Because the region of Pro161 is extending from the proteins core and displays high ps-ns flexibility.

It's worth noting that the observed disruption of a hydrogen bond in the trans form is a consequence of P161 existing in trans. The NMR data, which may lack trans-specific information, would naturally favor a structure with an intact hydrogen bond, as seen in the cis form.

We have revised the main text to clarify these potential limitations and ensure they are evident to the reader.

Minor Issues:

- In the introduction, the authors state: “In the folding state, the cis isomer is favored because it allows stronger stabilizing interactions compare to the trans isomer,...” (Line 41-42). I assume that this statement concerns P161 in the N2 domain? There is no reference to this protein in the preceding sentences, so that this statement can be taken as a general statement, concerning the behavior of prolines in any all proteins. I suppose it is not?

This statement is correct and refers to all protein containing a cis-Proline in the folded state. The cis isomer must allow more stabilizing interactions otherwise will stay in the trans-conformation.

- There is no error-bar in the bar-diagrams. This has few importance in the case of Figure 2 (expression yields), but in the case of thermodynamics parameters (Fig.3 and 4) or populations (Fig.5), this precludes the evaluation of the relevance of the data, especially when comparing the differences between different mutants. This is particularly the case for Figure 5 where, without error-bars, it is difficult to evaluate the significance of the slight variations of the cis- population between the different mutants.

We have added the error bars to Figure 3, 4, and 5.

- The authors gave few details on the NMR restraints (which residues are concerned by the measurement of RDCs? How H-bonds were determined? Etc...) and on the modeling of the protein (ARIA standard parameters is a little bit short...). I think that this should be developed in further details in the material & methods section.

We agree and have expanded the materials and methods section.

“H-bonds were introduced if all the following criteria were fulfilled: amide exchange of the corresponding amide is slowed down (Zierer et al, J Mol Biol, 2010), NOE patterns are in agreement with h-bonds, they are located in secondary structure elements confirmed by chemical shifts and initial structures.”

“RDCs were used for isolated amide signals with 1H15N NOEs > 0.6. RDCs are located all over the structure.”

“Structure calculations were performed using ARIA 2.3 [51]. 50 structures have been calculated using NOEs as ambiguous distance restraints, above mentioned Phi-Psi dihedral angle constraints, h-bonds and RDCs, and standard ARIA parameters for each Pro161 in cis or trans.”

- Supp. Figure 4. It is difficult to evaluate the quality of the experimental data from the overlays of the different chevron plots (with a “c” not a capital: chevron refers to a “geometric shape” not to M. Chevron). It should be better to keep this figure with only the fits, and to present independently the individual fits (with the experimental points) in additional figures (in the Sup. Mat.)

Thanks for the comment. We separated now Supp. Figure 4 in two Figures, Supp. Figure 4 and 5. In Supp. Figure 4 we show the kinetic raw data without chevron analysis and in Supp. Figure 5 with the respective fits.

Reviewer 2 Report

Comments and Suggestions for Authors

The authors study the effects of prolyl isomerization on protein folding, refolding, and stability using a hyperstable mutant of the N2 domain of the gene-3 protein from filamentous phage fd. This is a continuation of studies started over a decade ago when it was found that Pro161 preferentially adopts cis in the domain's native conformation; the unfolded state, of course, favors trans. Here, phi value analysis was used to map natively folded positions in the folding transition state, and NMR spectroscopy to model the differences in native structure when Pro161 is in cis as compared to trans. The results support the idea that native contacts in beta strands 4 and 5 provide the folding free energy to shift the cis/trans equilibrium of Pro161 toward cis. Overall, across multiple papers, the authors have provided a comprehensive analysis of the folding and stability of this protein domain, particularly focusing on the significance of prolyl isomerization at Pro161. The new results strengthen the overall body of work.   

The methods seem fine, and described appropriately. The authors have experience with the methods that were used. There might be some issues with the analysis, as detailed below with concerns #1 and #2. Addressing such issues might help to strengthen the paper.  

Concerns:  

1. The unfolding free energies, as measured from equilibrium methods, were modeled as two-state. Yet the authors propose a 4-state model for folding. Why would a two-state model be appropriate here? It seems this choice is owing to the vastly different time scales for refolding versus proline isomerization. However, it's not fully clear to me.  

2. I'm not sure if the good agreememt in the midpoints (line 265) confirms the applicability of the two-state model, especially if the heat-induced unfolded state was prone to aggregation suggesting that thermal unfolding wasn't measured under equilibrium conditions while the urea-induced unfolding was. Again, some justification for the two-state model would help.  

3. Two mutations, R142A and L198P, were unique in that they both strongly stabilized the native state, as compared to the other mutants that strongly destabilized the native state or had neglible stability effects. I would be interested in the authors thoughts on these two stabilizing mutations.  

4. Line 38, page 1, what exactly is meant by "temporarily halting folding and allowing conformational energy to accumulate"? If folding is halted, how could folding free energy accumulate? Rewriting this sentence for clarity could be helpful to readers. The following two sentences on lines 39 and 40 help, but the sentence on line 38 is confusing to me.  

5. Differing refolding kinetics suggests different folding transition states, correct? Could the authors elaborate on this?  

6. Supplementary Table 2 is mostly unreadable.  

7. Line 196, signifies is misspelled as "sgnifies”.

Author Response

#Reviewer2:

The authors study the effects of prolyl isomerization on protein folding, refolding, and stability using a hyperstable mutant of the N2 domain of the gene-3 protein from filamentous phage fd. This is a continuation of studies started over a decade ago when it was found that Pro161 preferentially adopts cis in the domain's native conformation; the unfolded state, of course, favors trans. Here, phi value analysis was used to map natively folded positions in the folding transition state, and NMR spectroscopy to model the differences in native structure when Pro161 is in cis as compared to trans. The results support the idea that native contacts in beta strands 4 and 5 provide the folding free energy to shift the cis/trans equilibrium of Pro161 toward cis. Overall, across multiple papers, the authors have provided a comprehensive analysis of the folding and stability of this protein domain, particularly focusing on the significance of prolyl isomerization at Pro161. The new results strengthen the overall body of work.
The methods seem fine, and described appropriately. The authors have experience with the methods that were used. There might be some issues with the analysis, as detailed below with concerns #1 and #2. Addressing such issues might help to strengthen the paper.
Concerns:

1. The unfolding free energies, as measured from equilibrium methods, were modeled as two-state. Yet the authors propose a 4-state model for folding. Why would a two-state model be appropriate here? It seems this choice is owing to the vastly different time scales for refolding versus proline isomerization. However, it's not fully clear to me.

If the cis/trans isomers (and any other conformers) within the folded ensemble can equilibrate quickly (relative to the timescale of the unfolding experiment), the system behaves as if there is a single “folded” population. Similarly, all unfolded species exist in a single “unfolded” population at equilibrium. For the urea-induced unfolding transitions we mixed the samples and incubated for 1h before measuring unfolding transition, enough time for equilibration. For thermal induced unfolding we used a slow (normal) heating ramp of 1K/min for equilibration (the cis/trans isomerization gets faster at higher temperatures).

We have previously shown (Jakob & Schmid, JMB, 2008, 1560-1575) that thermal and denaturant induced unfolding transitions of the N2 domain are reversible (measured with CD & fluorescence spectroscopy), displaying a single sigmoidal transiIon. No addiIonal plateaus or shoulders appeared in the transitions (which might suggest stable intermediates). As these criteria are met, under equilibrium conditions, a simplified two-state model (folded vs. unfolded) is justified (empirical confirmaIon).

2. I'm not sure if the good agreement in the midpoints (line 265) confirms the applicability of the two-state model, especially if the heat-induced unfolded state was prone to aggregation suggesting that thermal unfolding wasn't measured under equilibrium conditions while the urea-induced unfolding was. Again, some justification for the two-state model would help.
Thanks for this comment. You are right. Matching unfolding midpoints by itself does not rigorously prove a two-state mechanism. It is more of a supporting indicator that thermal and chemical denaturation are capturing the same principal unfolding process. Matching midpoints make it more likely that the protein transitions occur directly from folded to unfolded under both conditions and imply a single cooperative event governs both unfolding reactions. We have therefore reworded this paragraph, removing the statement of the two-state confirmation.

3. Two mutations, R142A and L198P, were unique in that they both strongly stabilized the native state, as compared to the other mutants that strongly destabilized the native state or had neglible stability effects. I would be interested in the authors thoughts on these two stabilizing mutations.
The surface exposed loop aa130-145 (including R142) is involved in binding to the F-Pilus, the natural receptor of the gene-3-protein on the E. coli surface (Filamentous bacteriophage: biology, phage display, and nanotechnology applications. Rakonjac, et al., Current Issues in Molecular Biology, 13(2), 51–76). The observed strong stabilization upon mutation to alanine suggests that the wild-type arginine was evolved for biological function and not for stability. L198 is located in a functionally non-relevant region of the N2 domain. There is a crystal structure of the full length gene-3-protein with the L198P mutation available (3DGS.pdb, https://doi.org/10.1016/j.jmb.2008.06.073) and comparison to the wild-type protein suggest a dual way of Pro198 stabilization: by decreasing the entropy of the unfolded state and by interacting favorably with Arg140 and Met135 in the folded state.

4. Line 38, page 1, what exactly is meant by "temporarily halting folding and allowing conformational energy to accumulate"? If folding is halted, how could folding free energy accumulate? Rewriting this sentence for clarity could be helpful to readers. The following two sentences on lines 39 and 40 help, but the sentence on line 38 is confusing to me.

The paragraph starts with cis-peptide bonds in native protein structures and that they have to undergo trans-to- cis isomerization during protein folding. The protein folding process starts with a trans-proline and then it stops because with a trans-Proline not all native protein contacts can be established. As long as the peptide bond is in a trans-conformation you have a kinetic barrier that prevents the completion of protein folding. The protein cannot progress past this point until the proline bond is isomerized from trans to cis. During this “pause,” the rest of the protein has already folded as much as it can, creating tension or “conformational energy.” Because the protein is stuck in a strained state, that built-up energy helps drive the slow proline isomerization step forward. Once the proline has isomerized, the last part of the protein can snap into its final conformation very quickly, completing the folding process.

We slightly reworded the paragraph to improve understandability.

5. Differing refolding kinetics suggests different folding transition states, correct? Could the authors elaborate on this?
Actually, No. The simplest explanation would be, that this single point mutation destabilizes the transition state, or in other words, as we typically mutate to Alanine, the removal of a side chain destabilizes the folding transition state, or in other words, in the wild-type protein this side chain was contributing to the folding transition state. In principle changed refolding kinetics can have three origins:

1. Increased energy barriers: The mutation introduces new kinetic barriers in the folding pathway, making the transition to the native state slower.
2. Altered folding intermediates: The mutation may create new folding intermediates or stabilize existing ones, causing the folding pathway to become more complex or slower.

3. Destabilization of the folding transition state: The mutation may destabilize the critical folding transition state, delaying progress toward the folded conformation.

In our case, we know that we have a well defined folding transition state for the N2’ domain comprising the two Beta-sheets left and right to Pro161, and here the observed destabilization of the transition state is the most plausible explanation.

6. Supplementary Table 2 is mostly unreadable.

We have increased the font size in Supplementary Table2 and have distributed this big table now over two pages.

7. Line 196, signifies is misspelled as "sgnifies”.

We corrected this error.

Round 2

Reviewer 1 Report

Comments and Suggestions for Authors

The authors answered to all my questions and/or gave convincing arguments. This manuscript can be published now as it is.